# Polyphenolic Maqui Extract as a Potential Nutraceutical to Treat TNBS-Induced Crohn’s Disease by the Regulation of Antioxidant and Anti-Inflammatory Pathways

**DOI:** 10.3390/nu12061752

**Published:** 2020-06-11

**Authors:** Tamara Ortiz, Federico Argüelles-Arias, Matilde Illanes, Josefa-María García-Montes, Elena Talero, Laura Macías-García, Ana Alcudia, Victoria Vázquez-Román, Virginia Motilva, Manuel De-Miguel

**Affiliations:** 1Department of Normal and Pathological Cytology and Histology, University of Seville, Avda. Sánchez-Pizjuán s/n, 41009 Sevilla, Spain; alamo110@us.es (M.I.); lmacias@us.es (L.M.-G.); mvazquez2@us.es (V.V.-R.); 2Department of Medicine, University of Seville, Avda. Sánchez-Pizjuán s/n, 41009 Sevilla, Spain; farguelles@telefonica.net (F.A.-A.); jfgarcia@us.es (J.-M.G.-M.); 3Department of Gastroenterology, University Hospital Virgen Macarena, c/Dr. Fedriani, nº 3, 41009 Sevilla, Spain; 4Department of Pharmacology, University of Seville, c/Prof García González, nº 2, 41012 Sevilla, Spain; etalero@us.es (E.T.); motilva@us.es (V.M.); 5Department of Organic and Pharmaceutical Chemistry, University of Seville, c/Prof García González, nº 2, 41012 Sevilla, Spain; aalcudia@us.es

**Keywords:** maqui, polyphenols, Crohn’s disease, inflammatory bowel disease, acute inflammation, macrophages, antioxidants, inflammatory proteins

## Abstract

Nutraceuticals include a wide variety of bioactive compounds, such as polyphenols, which have been highlighted for their remarkable health benefits. Specially, maqui berries have shown great antioxidant activity and anti-inflammatory effects on some inflammatory diseases. The objectives of the present study were to explore the therapeutic effects of maqui berries on acute-phase inflammation in Crohn’s disease. Balb/c mice were exposed to 2,4,6-trinitrobenzene sulfonic acid (TNBS) via intracolonic administration. Polyphenolic maqui extract (Ach) was administered orally daily for 4 days after TNBS induction (Curative Group), and for 7 days prior to the TNBS induction until sacrifice (Preventive Group). Our results showed that both preventive and curative Ach administration inhibited body weight loss and colon shortening, and attenuated the macroscopic and microscopic damage signs, as well as significantly reducing transmural inflammation and boosting the recovery of the mucosal architecture and its muco-secretory function. Additionally, Ach promotes macrophage polarization to the M2 phenotype and was capable of down-regulating significantly the expression of inflammatory proteins COX-2 and iNOS, and at the same time it regulates the antioxidant Nrf-2/HO-1 pathway. In conclusion, this is the first study in which it is demonstrated that the properties of Ach as could be used as a preventive and curative treatment in Crohn’s disease.

## 1. Introduction

Inflammatory bowel diseases (IBDs) are chronic idiopathic disorders of the gastrointestinal tract that mainly include two types: ulcerative colitis (UC) and Crohn’s disease (CD) [1]. Their prevalence has rapidly increased in the last decade, affecting five million patients worldwide, with the highest incidence in Northern Europe and Northern America [2]. Although the etiology is still unclear, it might be caused by various factors that could play a key role in the onset and progression of IBD, such as an exaggerated immune response to environmental factors, including the composition of the luminal microbiota in genetically susceptible individuals [3]. Particularly, CD is a complex disorder due to its variable locations, heterogeneous pattern of disease severity and behaviour, among others. A variety of proinflammatory mediators, including cytokines, oxidative stress (OS), disruption of the intestinal epithelial barrier and transmural inflammation can be seen in this illness. It can affect any portion of the digestive tract, although it commonly affects the ileum and the colon, and is marked by periods of exacerbation and remission, resulting in abdominal pain, diarrhea, weight loss and a generally poor quality of life [4]. The immune mechanism underlying CD pathogenesis is an aberrant adaptive immunity mediated by CD4+ helper T (Th) cells, specifically by Th17 and Th1, with an over-expression of IFN-γ, TNF-α, IL-2 and IL-12 production, together with suppression of the activity of T regulatory cells (Treg) [5,6]. Macrophages are an essential part of the innate immune system and provide important protection against harmful local antigens, such as those that cause intestinal inflammation. In tissues, macrophages are activated and produce multiple cytokines in response to various signals, and change to classical M1 (pro-inflammatory) or M2 (anti-inflammatory) phenotypes. Upon stimulation, M1 phenotypes produce high levels of cytokines, such as TNF-α, IL-1β, IL-12, IL-18 and IL-23, chemokines (CXCL9, CXCL10), reactive oxygen species (ROS) and reactive nitrogen species (RNS), and are involved in driving Th1- and Th17- mediated immune responses. M2 macrophages, also referred to as ‘alternatively activated’, are marked by the expression of arginase-1 (ARG-1), IL-10 and IL-13, and have an immune-regulatory function [7,8]. The balance between inflammatory M1 and anti-inflammatory M2 cells could determinate the disease’s progress, and therefore the factors implicated in the disruption of the balance toward an increase of the M2 macrophages cells could offer unique approaches for the future management of IBD [9].

The onset and progress of inflammation is directly related to two enzymes, cyclooxygenase-2 (COX-2) and inducible nitric oxide synthase (iNOS), associated with an increment of inflammatory lesions in the intestinal tissue [10]. In patients with CD, the higher immunoexpression of COX-2 in epithelial and inflammatory cells of the lamina propria has been observed [11], whereas in the inflamed gut of active ileocecal or colonic CD, iNOS is massively accumulated in subepithelial areas [12].

Interestingly, hemoxygenase-1 (HO-1) is one of the most important mechanisms of antioxidant defense, and its transcription depends on the nuclear erythroid 2-related factor 2 (Nrf-2), which is sequestered in the cytoplasm by the actin-binding protein Keap1, and it is responsible for the regulation of cellular redox balance. The presence of OS leads to the phosphorylation of serine/threonine residues in Nrf-2, to dissociate from Keap1 and translocation of Nrf-2 to the nucleus, and promotes binding to a specific DNA sequence known as the Antioxidant Response Element (ARE), resulting in an up-regulation of HO-1 [13]. Moreover, an over-expression of Nrf-2 has been observed in acute experimental models of colitis [14] and biopsies of inflamed tissue from patients with gastritis and IBD [15], leading to better inflammation regulation.

Furthermore, experimental colitis induced by hapten reagent 2,4,6-trinitrobenzene sulfonic acid (TNBS) is characterized by a predominant Th1/Th17-mediated immune response with elevated production of IL-12, IL-17, IL-18, IL-23, IL-27 and IFN-γ, and mucosal inflammation, which closely resembles important immunological and histopathological aspects of human CD [16,17]. In particular, the use of animal models is necessary to develop new therapies, since the conventional treatments for IBD (e.g., corticosteroids and immunomodulators) and biological therapies have many side effects and relative responses [18]. In this sense, nutraceuticals, including bioactive compounds such as polyphenols from berries with anti-inflammatory and antioxidant activities, could be potential agents for the treatment of IBD in humans [19]. Amongst them, *Aristotelia chilensis* (Mol) Stuntz (*Elaeocarpaceae)*, commonly known as maqui, is an edible black-colored fruit and endemic Chilean berry which has an exceptionally high content of phenolic compounds with high antioxidant capacity [20] and, less studied, anti-inflammatory effects [21]. In a previous study, using a cellular model of OS consisting of hydrogen peroxide-treated (H_2_O_2_) HT-29 colon epithelial cells, we showed that our polyphenolic maqui lyophilized extract (Ach from now on) was rich in polyphenols and had a significant antioxidant effect, as evaluated through the intracellular ROS concentration using a DCFH-DA assay, protecting the intestinal epithelial cells against oxidation (unpublished).

The goal of the present work was to assess the effects of Ach on clinical, histological and inflammatory parameters using an animal model of acute phase of CD. We have analysed the preventive and curative effects of Ach during early colonic inflammation in mice caused by TNBS. Specifically, we have investigated the effect of Ach through macroscopy and histological study, as well as examining the implications for macrophage frequency and phenotype (M1 and M2). Furthermore, we determined, by Western blotting, the anti-inflammatory and antioxidant mechanisms through the expression of iNOS and COX-2, and the HO-1/Nrf-2 signaling pathway, respectively, in the colon mucosa of mice.

## 2. Materials and Methods

### 2.1. Extract

In order to obtain an extract including total polyphenols, the extraction was performed using the acid MeOH, according to the method published by Genskowsky et al. [22], with slight modifications. The initial sample consisted of 50 g lyophilized powdered wild maqui fruit (seed and pulp) from a packaged and commercialized product (Isla Natura de Chile^®^, Chiloé, Chile). In total, 250 mL of MeOH/H^+^ (0.1% HCl) at pH 1 were added to the sample, and then homogenized with an ultrasound device (Hielscher Ultrasound Technology UP400S, Teltow, Germany). The extract was centrifuged at 4000 rpm for 10 min at 4 °C, and the supernatant was collected. This procedure was repeated 5 times. The supernatants were mixed in a round-bottomed flask and evaporated until dryness using a rotary evaporator (Büchi B-490, Hampton, VA, USA). The dried extract was dissolved in distilled water and centrifuged at 4000 rpm for 4 min. Finally, it was passed through 2 filtering processes (100–150 MM and 40–100 MM) and then lyophilized in a Telstar Cryodos Freeze Dryer (Tokyo, Japan). The final extract was stored at −20 °C. All experiments were carried out under darkness and controlled temperature.

### 2.2. Animals

Male Balb/c mice aged 12–14 weeks old, weighing 19–35 g, were kept in pathogen-free cages and controlled laboratory conditions (temperature 20 ± 2 °C, humidity 40–50%, lighting regimen of 12 light/12 dark hours). Animals were fed a normal laboratory diet (Teklad diet, Envigo, Cambridgeshire, UK) and fresh tap water, ad libitum. The study was approved by the Ethical Committee of the Faculty of Medicine, University of Seville (name of the institution: ‘Consejería de agricultura, pesca y desarrollo rural, Junta de Andalucía’; approval date: 19 January 2017; approval number: 03/03/2017/032). Animal handling was conducted in accordance with the Guide for the Care and Use of Laboratory Animals [23].

### 2.3. Induction of Colitis and Treatment

Animals were randomly assigned to 4 groups (*n* = 6). After fasting for 12 h, animals were anesthetized with ventilatory anesthesia and CD was induced according to the method reported by Liu T.J. et al. [24] with modifications. A single dose of TNBS at a concentration of 100 mg/kg was dissolved in ethanol (EtOH 50%) and installed intra-rectally, using at a total volume of 70 µL. The EtOH of this solution acts not only as the vehicle but also breaks the mucosal barrier and TNBS haptenizes colonic proteins, turning them immunogenic to the host’s immune system [17]. Polyphenolic maqui extract (Ach) was administered daily, with a single dose of oral gavage of Ach at 50 mg/kg/day, for 4 days after TNBS administration (Curative Group) and for 1 week prior to the induction of the disease, until sacrifice (Preventive Group) (Figure 1).

### 2.4. Macroscopic Evaluation of Colonic Damage

Mice were sacrificed 5 days after the onset of the experiment under intraperitoneal anesthesia. Once the death of the animals was confirmed by the absence of a response to a toe pinch and by touching the cornea, the abdominal cavity was opened, the entire large intestine was removed and lightly cleaned using physiological saline to remove fecal residues, and then was measured. Images of the colonic morphology were captured using a Canon EOS 350 zoom camera (Canon Inc., Tokyo, Japan). Thereafter, the intestine was opened longitudinally, and the macroscopic damage score was assessed by an independent observer, who was unaware of the groups’ code. Damage was scored according to a modified version of the Appleyard and Wallace score [25]. Alterations in the intestinal mucosa were scored on a 0 to 11 scale, and the following items were considered: absence of damage, focal or little hyperemia without ulceration, bowel wall thickening, ulceration and local inflammation, two or more ulcerated and inflamed areas extending >1 cm along the length of the colon, damage extending >2 cm in length, one point for each cm from 2 cm on the damaged area, mucus and diarrhea (absent or present), intestinal adhesions and congestion (0–2 score). The other longitudinal half of the colon was collected and frozen in liquid nitrogen for later analysis.

### 2.5. Histopathological Study

For the histopathological study, half of the entire length of the large intestine was rolled up from the distal to the proximal end in order to evaluate the whole organ and its characteristics in only one slide. The longitudinal fractions of the colon from different groups were harvested and fixed overnight with 4% buffered paraformaldehyde and embedded in paraffin. Thereafter, sections of tissue were cut at 4 µm on a rotary microtome (Microm HM 310, Thermo Scientific, MA, USA), mounted on glass slides and dried for 2 h at 60 °C before staining with different methods. All histology slides were blindly analysed by two pathologists using an Olympus microscope (Vanox AHBT3, Tokyo, Japan).

#### 2.5.1. Hematoxylin & Eosin Staining

The tissue sections were deparaffinized, hydrated and stained with hematoxylin–eosin, according to standard protocols. All histology slides were examined for histological evaluation of colonic inflammation and characterization of histopathological changes. Analysis was performed to establish a microscopic score system based on one which was previously described, but with some modifications [25,26]. Criteria include inflammatory infiltrate, goblet cell loss, crypt density, submucosal infiltration (all categorized from 0–3, corresponding to absent, mild, medium and severe) and ulcerations, crypt abscesses and necrosis (scored 0 or 1, corresponding to absent or present).

#### 2.5.2. Periodic Acid Schiff Staining

A periodic Acid Schiff (PAS) stain was used to evaluate mucosal mucin production and the presence of the goblet cells. Briefly, the tissue sections were treated with periodic acid at 0.5% for 5 min and washed with distilled water. Finally, they were treated with the Schiff’s reactive for 20 min, and again washed and stained with Harris hematoxylin. The goblet cells were assessed in the whole colon tissue at a resolution of 1× and 10×.

### 2.6. Macrophage Immuno Histochemistry

Immunostaining was performed on a Ventana Bench Mark GX stainer (Ventana Medical Systems, Roche Group, Tucson, AZ, USA) with pre-diluted Ventana anti-CD-68, as a marker of M1 macrophages, and anti-CD163, as a marker of M2 macrophages; both rabbit monoclonal primary antibody, and the Ventana Optiview DAB IHC detection kit—according to the manufacturer’s instructions—were used (Ventana Medical Systems). Nuclei were counterstained with hematoxylin. Positive cells from samples of each animal were quantified and averaged from ten fields at high magnification (40×).

### 2.7. Isolation of Cytoplasmic Proteins and Western Blot Assay

Frozen colonic tissues were randomly selected (4 per group), weighted and homogenized in ice-cold buffer (50 mM Tris-HCl, pH 7.5, 8 mM MgCl_2_, 5 mM ethylene glycol bis (2-aminoethyl ether)-N,N,N′,N′-tetra acetic acid, 0.5 mM EDTA, 0.01 mg/mL leupeptin, 0.01 mg/mL pepstatin, 0.01 mg/mL aprotinin, 1 mM phenylmethylsulfonyl fluoride (PMSF) and 250 mM NaCl) in a proportion of 1:3. The homogenates were centrifuged at 12,000× *g*, 4 °C for 15 min, and the supernatants were collected and stored at −80 °C. The protein concentration of the colon homogenates was determined following the Bradford colorimetric method [27]. Aliquots of supernatants containing equal amounts of protein (30 μg) were separated on 10% acrylamide gel by sodium dodecyl sulfate polyacrylamide gel electrophoresis. In the next step, the proteins were electrophoretically transferred onto a nitrocellulose membrane, stained with Ponceau red to check transfer problems and the equal loading of total proteins, and blocked with 5% BSA in Nonidet^TM^ at 0.5% and PBS. Then, they were incubated with specific primary antibodies: rabbit anti-inducible nitric oxide synthase (iNOS) (1:1000; Stressgen-Enzo Life Sciences, Farmingdale, NY, USA), rabbit anti-COX-2 (1:3000; Cayman Chemical, Ann Arbor, MI, USA), rabbit anti-Nrf2 (1:1000; Santa Cruz, Texas, USA) and rabbit anti-HO-1 (1:1000; Stressgen-Enzo Life Sciences, Farmingdale, NY, USA) overnight at 4 °C. After rinsing, the membranes were incubated with the horseradish peroxidase-linked (HRP) secondary antibody anti-rabbit (1:1000; Cayman Chemical, Ann Arbor, MI, USA) or anti-mouse (1:1000; Dako, Atlanta, GA, USA) containing blocking solution for 1–2 h at room temperature. To prove equal loading, the blots were analyzed for β-actin expression using an anti-β-actin antibody (1:1000; Sigma Aldrich, St. Louis, MO, USA). Immunodetection was performed using an enhanced chemiluminescence light-detecting kit (SuperSignal West Pico Chemiluminescent Substrate, Pierce, IL, USA). Then, the immunosignals were monitored using an Amersham imager 600 (Healthcare Life Sciences, Buckinghamshire, UK), and densitometric data were studied following normalization to the housekeeping loading control. The signals were analyzed and quantified using Image Processing and Analysis in Java (Image J, Softonic, National Institute of Mental Health, Bethesda, MD, USA), and expressed as percentages in respect to the Control Group.

### 2.8. Statistical Analysis

All values are expressed as arithmetic means ± standard error of the mean (SEM). Heterogeneity was tested by using Levene’s test and the Shapiro–Wilk test for normality. When the variables were normally distributed, Student’s t-test was used along with the Mann–Whitney test for non-normally distributed data for the comparison of 2 means. For the comparison of 3 means, the parametric value groups were analyzed by one-way analysis of variance (ANOVA), or Kruskal–Wallis for the non-parametric values, followed by Bonferroni’s post hoc. *p*-values of <0.05, <0.01 or <0.001 were considered statistically significant. Statistical analysis and comparisons among means were carried out using STATA software (version 12, 2011, StataCorp, TX, USA).

## 3. Results

### 3.1. Therapeutic Effects on Macroscopic Damage of Ach on TNBS-Induced Crohn’s Disease

Intracolonic administration of TNBS successfully induced a decreased stool consistency, rectal bleeding and significative weight loss in untreated mice. During the experiment, mice from the CD group (TNBS+EtOH 50%) showed a remarkable weight loss at 24 h after colitis induction which continued with a downward trend until sacrifice, compared to the Control Group (EtOH 50%). When 50 mg/kg of Ach was administered by an orogastric tube, the Curative and Preventive Groups showed a slight loss of weight the day after induction, which was recovered during the experiment period (Figure 2A). At the end of the experiment, significant differences were observed in the length of the colon. TNBS-induced CD mice exhibited a considerably shortened large intestine (*p* < 0.001), as an indicator of tissue inflammation, compared to the Control Group. On the contrary, the administration of Ach in the Curative and Preventive Groups significantly restored the colon length (*p <* 0.05 and *p <* 0.01, respectively) (Figure 2B–C). In line with the clinical findings, macroscopic examination of the colonic mucosa of the TNBS-induced CD mice evidenced hyperemia, congestion and ulceration, accompanied with mucus and liquid stool inside the colon. Additionally, a greater number of adhesions of the colon to the adjacent organs in the CD Group were observed. Colonic morphology and macroscopic characteristics were significatively restored in all groups after treatment with Ach (*p <* 0.001) (Figure 2D).

### 3.2. Histological Effects of Ach on TNBS-Induced Crohn’s Disease

The histopathological evaluation showed a transmural inflammation distributed in areas characterized by the loss of histological structure and a massive infiltration of inflammatory cells, specially neutrophils and lymphocytes associated with acute phase of inflammation, when compared with the Control Group. Mucosal and submucosal ulcerations, necrosis and edema in lamina propria were observed throughout the whole colon of the CD Group. After the administration of Ach (Curative and Preventive Group), the intestinal lesions revealed a reduction in inflammation, significant recovery of the pathological alterations, and the normal histological structure of the colon was re-established (Figure 3A–D). The histological score of the large intestine showed a significant increase (*p <* 0.001) in tissue damage in the untreated CD Group compared to the Control Group. Ach significantly reduced the histopathological score (*p <* 0.001) (Figure 3E), suggesting a protective and curative effect in the acute inflammation of CD. We also assessed PAS staining on colon tissue to determine the distribution of goblet cells and the presence of glycoproteins, including mucin content, within the colonic mucosa. TNBS-induced CD showed a negative staining for PAS (PAS-), indicating a significative reduction in the amount of goblet cells and mucin content. In contrast, the administration of Ach in the Preventive and Curative Group exhibited a significant restoral to a normal pattern in colonic tissue form, similarly to Control Group, evidenced by the positive staining for PAS (PAS+) (Figure 4A–D).

### 3.3. Impact of Ach on the Polarization to M1 and M2 Macrophages

Macrophage subtypes in the intestinal sections were visualized using immunohistochemistry. CD68+ (M1) and CD163+ (M2) macrophage cells were found within the layer of the lamina propria and submucosa in all groups. The amount of CD68+ cells was significantly higher (*p <* 0.001) in the CD Group compared with the Control Group, showing the activation of the innate immune response and the onset of the inflammatory process in CD. However, the Ach administration was not able to reduce the M1 macrophage expression, although we found a significative difference between the Curative and the Preventive Group (*p <* 0.05) (Figure 5A,B)**.** It has been well established that M2 macrophages exert a modulatory role in TNBS-induced colitis [16]. Consequently, fewer M2 macrophages were expected to be found in the colonic tissue of TNBS-induced CD mice. On the other hand, we observed a significative increase in M2 cells in Ach treated mice (Curative and Preventive Group: *p* < 0.01 and *p* < 0.001, respectively) compared to the non-treated CD Group. Similar values were found in the Control Group, revealing a constant population of regulatory M2 macrophages in the colon (Figure 5C,D)**.**

### 3.4. Anti-Inflammatory Effect of Ach on the Protein Expression of iNOS and COX-2 Measured by Western Blot

To explore the possible mechanisms of the protective effect of Ach observed in the morphological and histopathological studies, we examined the expression levels of different inflammatory proteins in colonic mucosa. Western blot analysis showed that the TNBS-induced CD led to a significant increase in iNOS expression (*p* < 0.05), a pro-inflammatory enzyme related to the onset and perpetuation of inflammation [28]. However, the curative treatment with Ach after the induction with TNBS resulted in a significant down regulation of iNOS expression levels in comparison with the untreated group, while the Preventive Group showed a notable decrease in the iNOS expression (Figure 6A,B). As regards the COX-2 protein, widely known for acute inflammatory activity in epithelial and inflammatory cells [29], we observed an up-regulated expression trend in the CD Group. A significant decrease in COX-2 expression (*p* < 0.05) in the Curative and Preventive Groups was observed when compared with the CD Group (Figure 6A–C).

### 3.5. Antioxidant Effect of Ach on Protein Expression of the Nrf-2/HO-1 Pathway, Measured by Western Blot

Based on the known antioxidant effect of Ach, Nrf-2/HO-1 pathway expression in the intestinal tissues was tested. As shown in Figure 7A,B, the level of Nrf-2 showed an over-regulation in the CD Group compared to the Control Group, without presenting significant differences. The Ach administration, in both the Preventive and Curative Groups, showed a slight upward trend in Nrf-2 protein expression. Likewise, HO-1 protein expression increased in TNBS-induced CD mice, with statistically significant values (*p <* 0.05). In line with Nrf-2 observations, HO-1 showed an increase when mice were given Ach, although this difference was not statistically significant (Figure 7A–C).

## 4. Discussion

CD is characterized by chronic transmural and segmental intestinal inflammation that can affect any part of the gastrointestinal tract. The etiopathogenesis is complex and multifactorial, and is not totally clear yet [30]. Patients typically experience periods of flare-ups and symptomatic remission, which are difficult to predict and treat adequately [31]. Additionally, the standard medical therapy presents a relative efficacy, with a significant number of patients that fail or lose response during therapy and present potentially serious side-effects [18,32]. Due to the complexity of CD and the multitude of treatments, new therapeutic strategies, both effective and safe for IBD, have been reported. From this perspective, the use of nutraceuticals, including bioactive compounds such as polyphenols, has been gaining interest recently [33,34]. Maqui, a black-coloured edible berry, endemic to Chilean Patagonia, has been used in traditional medicine to treat various digestive disorders [35]. Indeed, this berry has been described as containing the highest antioxidant powers, with an extraordinary content of polyphenols, mainly delphinidin, and relevant anti-inflammatory properties [20,36]. However, the maqui berry’s properties in the context of CD have not been researched yet. Regarding this, our extract has a total polyphenolic content of 39.02 mg/g (expressed as gallic acid), and we have found in previous in vitro experiments that polyphenolic maqui extract possesses powerful antioxidant effects in the H_2_O_2_-induced HT-29 epithelial cells (non published). Based on this background information, the aim of this study was to assess the preventive and curative effects of the administration of Ach on the development of the disease in the acute animal model of CD.

In our study, the intracolonic administration of 100 mg/kg of TNBS together with EtOH at 50% in a single dose was successful to induce CD in mice. The aim of the experimental parameters chosen, such as dosage and exposure to TNBS, was to achieve acute inflammation and avoid a high mortality rate, which is frequently associated with higher concentrations than 150 mg/kg [37]. Although TNBS colitis does not reproduce perfectly the etiopathogenesis of CD, our results showed the development of multifocal and transmural inflammation, ulcerations, fibrosis and a loss of goblet cells. Previous articles have evaluated the effect of TNBS on the induction of colitis in experimental models, demonstrating its similarity to human CD [24,38,39]. Nevertheless, the presence of epithelioid granuloma, a specific histologic feature of CD, was not found. The etiology and significance of granuloma in CD are still unclear, but it is only seen in less than half of the cases of CD, and it could be associated with chronicity, according to other authors [5]. Additionally, the treatment dose with Ach was chosen according to previously published data in relation to polyphenols from natural sources and treatments in the intestinal affections of experimental animal [40,41]. However, most studies showed positive effects reducing inflammation and clinical parameters using higher concentrations than our study [42,43].

The results of this present study reveal, for the first time, the protective and curative effect of the administration of Ach in an experimental model of TNBS-induced CD. The administration of 50 mg/kg of Ach enhanced clinical parameters (diarrhea and blooding), ameliorated body weight loss and was able to improve macroscopic characteristics and tissue inflammation markers, such as the length of the large intestine, in TNBS- exposed mice. The equivalent amount of fresh maqui fruit for the dose administered to the mice was 100 mg/day. This value, translated to a standard human weight, corresponds to 240 g of fresh fruit per day.

Consistent with our findings, Scarano et al. [44] reported that some foods enriched with different polyphenols were able to reduce clinical damage and disease severity in an experimental colitis model, while other total polyphenolic extracts, derived from natural sources, have shown improvements in clinical parameters like weight and macroscopic damage, which were achieved at concentrations of 200 mg/kg of extract [42], which represented a higher quantity than that administered in our study. In the histopathological analysis of our study, the administration of Ach influenced the significant reduction of TNBS damage in the colon epithelial tissue of acute CD, and decreased the infiltration of inflammatory cells into the mucosa and adjacent layers, which have been suggested to contribute significantly to mucosal dysfunction and necrosis of intestinal wall tissue [45]. The protective effect of polyphenols on the architecture of the crypts and the production of mucus in goblet cells has previously been reported [46]. We demonstrated that the oral administration of Ach significatively recovered the mucosal architecture and its muco-secretory function, as evidenced by the PAS positive cells in the colonic mucosa.

A proinflammatory phenotype of M1 macrophages supporting Th1 cells contributes to the secretion of proinflammatory cytokines, as well as ROS and RNS. Alternatively, activated or M2 macrophages, which are anti-inflammatory and immunoregulatory, are polarized by Th2 cytokines and produce anti-inflammatory cytokines [47]. It has been previously reported that polyphenols exert an anti-inflammatory action by modulating macrophage phenotypes in inflammatory diseases [48,49], but not in IBD as yet. Additionally, the effect of other bioactive compounds was also reported to induce the polarization of M1 towards M2 macrophages, reducing systemic cytokines and attenuating colitis symptoms [8,50]. The local impact of Ach on macrophage differentiation was also studied in this work. We specifically demonstrated that macrophages responded to TNBS upregulating M1 and downregulating M2 macrophages. When a rich polyphenol source like Ach was administrated, the stronger effect was observed in M2 macrophages with high immunoexpression, attenuating or preventing the development of TNBS-induced CD. These results could indicate that M2 macrophages were more susceptible to modulation after Ach treatment, enhancing the anti-inflammatory effects of this treatment. A significative reduction of M1 could probably be achieved in a chronic model of CD. In support of this finding, it has been reported that the lamina propria monocytes and M1 macrophages contribute to the disruption of the epithelial barrier, leading to chronic intestinal inflammation in patients with IBD [12].

Moreover, other mechanisms described in IBD are mainly related to inhibiting the expression of inflammatory proteins, such as iNOS and COX-2 [51]. Nitric oxide (NO•), a free radical, is generated from the oxidation of L-arginine by its three enzymatic sources, which have been localized in the gastrointestinal tract. However, the large amounts of NO• generated from iNOS have been implicated in tissue damage and intestinal inflammation [52]. On the other hand, evidence from experimental models of intestinal inflammation demonstrate an up-regulation of iNOS expression [10,53]. In the present research, we observed a significative decrease in the expression of iNOS protein in the Preventive and Curative treatment groups with Ach. Our results are in agreement with previous in vitro studies that showed the capacity of maqui to modulate inflammatory response through inhibition of iNOS expression, with the consequent decrease of NO• levels [21,36]. The COX-2 enzyme catalyzes the formation of pro-inflammatory prostaglandins, thromboxane and levuloglandinas that are directly associated with inflammatory lesions in the colon of murine model CD [10,46]. The relation between COX-2 immuno-expression in inflammatory cells of patients with CD and the pathogenesis of CD has been reported [11]. In this sense, the induction of CD by TNBS induces acute inflammation symptoms that are accompanied by COX-2 colonic over-expression in relation with the Control Group. Our data evidenced a reduced COX-2 expression in mice treated with preventive and curative oral Ach. These findings, at least in part, suggest that Ach may be involved in the suppression of iNOS and COX-2, and could be considered for the treatment of IBD, specifically in CD. In fact, a recent study has identified that the ethyl acetate fraction of maqui berry water extract considerably reduces the expression of COX-2 and iNOS in DSS-induced UC [54].

During the inflammatory processes, the OS plays an important role in promoting its maintenance throughout time. In this condition, there are well-established increases of Nrf-2 expression and the consecutive regulation of the expression of numerous cytoprotective genes encoding antioxidant enzymes, such as ferritin, superoxide dismutase (SOD), peroxiredoxin-1 (PRDX1) and gluthathione S-tranferases (GSTs) [13,55], and also its downstream target, HO-1, to ameliorate the oxidative effects [56,57]. In our study, an Nrf-2 and HO-1 up-regulation was observed in the CD Group, indicating the primary events of acute inflammation, probably as a compensatory system to alleviate inflammation and tissue damage. On the contrary, Ávila-Román et al. [57] demonstrated reduced Nrf-2 and HO-1 expression levels in colon samples from a TNBS chronic colitis model. In addition, several dietary phytochemicals, such as green tea polyphenol components [58], anthocyanins [59] and carotenoids [56], have been implicated in reversing inflammatory processes by increasing the expression of Nrf-2 and HO-1. Our research has revealed that Ach administration generated an upward trend in this antioxidant pathway, which may prevent ROS production. It is also possible that these effects may be significant in the long term, specifically in a recurrent model, and other antioxidant enzymes regulated by Nrf-2 could be involved during the first phase of oxidative and inflammation injury. Moreover, a recent relevant report showed that maqui extract could exert an important effect on the microbiota imbalance in IBD [54]. Our findings propose to consider Ach as a potential nutraceutical for preventive or curative treatment in acute phase inflammation in CD, modulating the oxidative status and inflammatory response.

## 5. Conclusions

In summary, for the first time, it has been demonstrated that mice that have received Ach preventively, once the disease has started, are protected from TNBS-induced CD. The oral administration of Ach prevents weight loss and drastically reduces colonic tissue damage. Furthermore, we have demonstrated a reduction of transmural inflammation, and an increase in mucosal architecture recovery and its muco-secretory function. It is likely that Ach mediates this anti-inflammatory activity through multiple pathways. Our study has provided evidence that Ach promotes macrophage polarization to the M2 phenotype. In addition, we observed that Ach achieves protective effects against intestinal damage induced by TNBS through reduction of pro-inflammatory proteins levels such as COX-2 and iNOS, as well as regulation of antioxidant responses through the Nrf-2/HO-1 pathway. These results indicate that Ach could be a novel approach to prevent and modulate the inflammatory response in IBD, especially in CD. Consequently, the clinical translation of these findings should be evaluated in the future.

## Figures and Tables

**Figure 1 nutrients-12-01752-f001:**
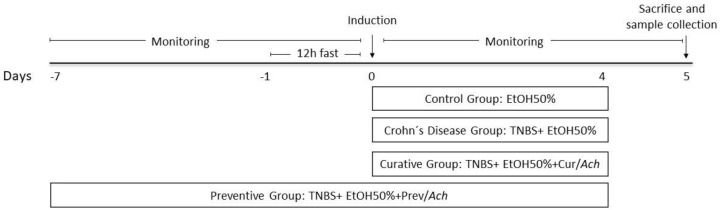
Schematic representation of experimental Crohn’s disease (CD) protocols and treatments with polyphenolic maqui extract (Ach). 100 mg/kg of TNBS plus EtOH 50% was administrated to induce CD (CD Group). Mice were treated with 50 mg/kg/day of Ach 4 days after CD induction (Curative Group), and 7 days before and 4 days after induction (Preventive Group). The whole colon was collected at 5 days after colitis induction for determination of the microscopic score, macroscopic damage, macrophage polarization, and inflammatory and antioxidant pathway activation. Monitoring of weight and clinical parameters was performed during the experiment.

**Figure 2 nutrients-12-01752-f002:**
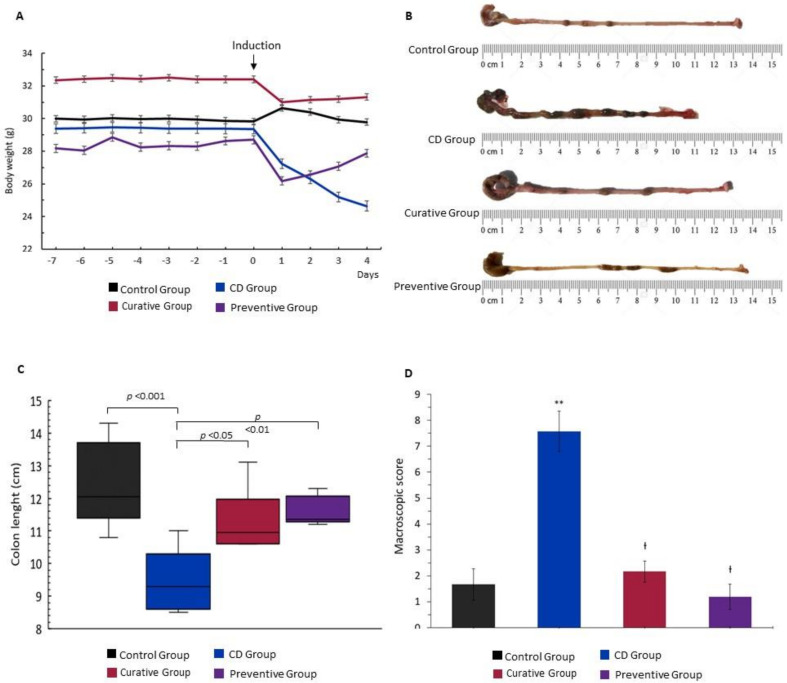
Polyphenolic maqui extract (Ach) prevents weight loss and ameliorates the macroscopic damage induced by TNBS in colon with Crohn’s disease (CD). (**A**) Body weight changes during the experiment. (**B**) Representative view of large intestine from the Control Group, CD Group, Curative Group and Preventive Group. (**C**) Colon length of the different experimental groups. (**D**) Colonic macroscopic score. Values represent mean ± SEM; ** Statistical significance *p* < 0.001 compared to Control Group; Ɨ *p* < 0.001 compared to CD Group.

**Figure 3 nutrients-12-01752-f003:**
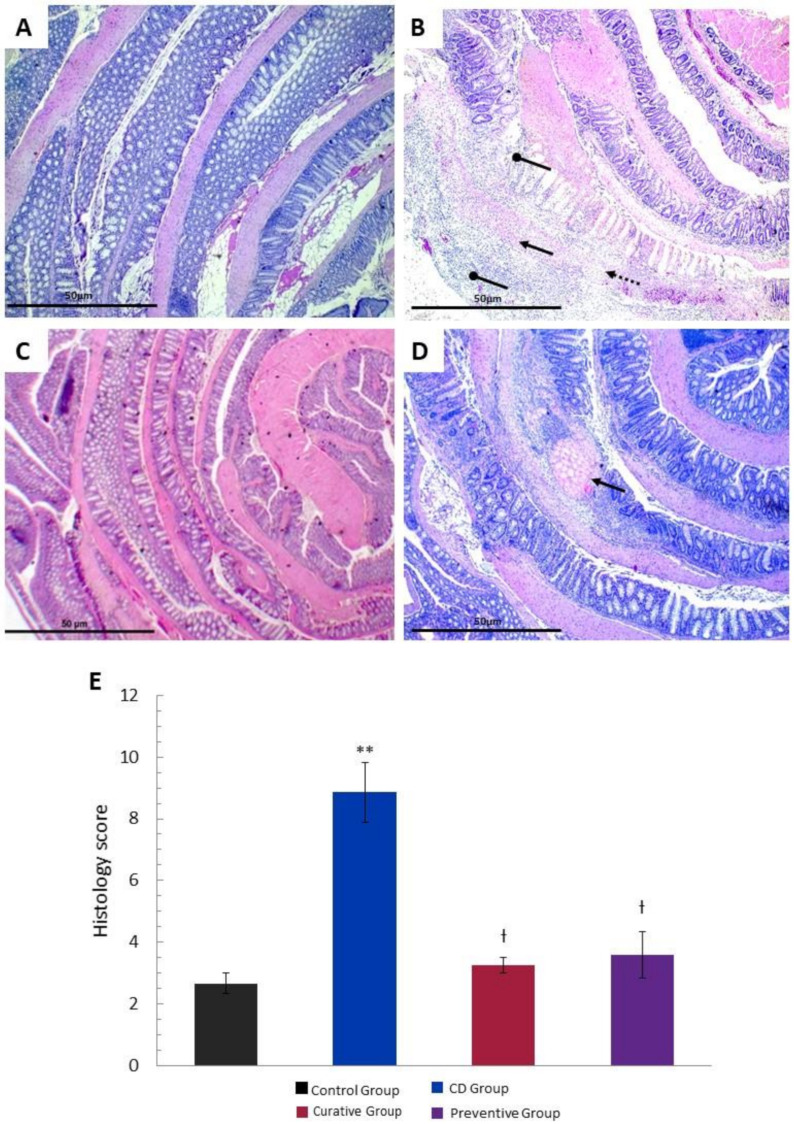
Preventive and curative effect of polyphenolic maqui extract (Ach) on microscopic evaluation and histopathological study of colonic tissue in a TNBS-induced acute Crohn’s disease (CD) model. Representative hematoxylin/eosin (H&E) staining of the rolled entire large intestine (4× magnification). (**A**) Intact colonic tissue of the Control Group. (**B**) Severe loss of glandular architecture distributed in foci (rounded arrow), mixed inflammatory infiltrate (arrow; neutrophils and lymphocytes) and edema (dashed arrow) in the CD Group. (**C**) Recovered histological structure with mild inflammation in the lamina propria in the colon of the Ach-treated Curative Group. (**D**) Preserved architecture throughout the entire colonic tissue in the Ach-treated Preventive Group, except for a single focus of necrosis (arrow). (**E**) Histology score of large intestines from all groups. Values represent mean ± SEM** Statistical significance *p* < 0.001 compared to the Control Group; Ɨ *p* < 0.001 compared to the CD Group.

**Figure 4 nutrients-12-01752-f004:**
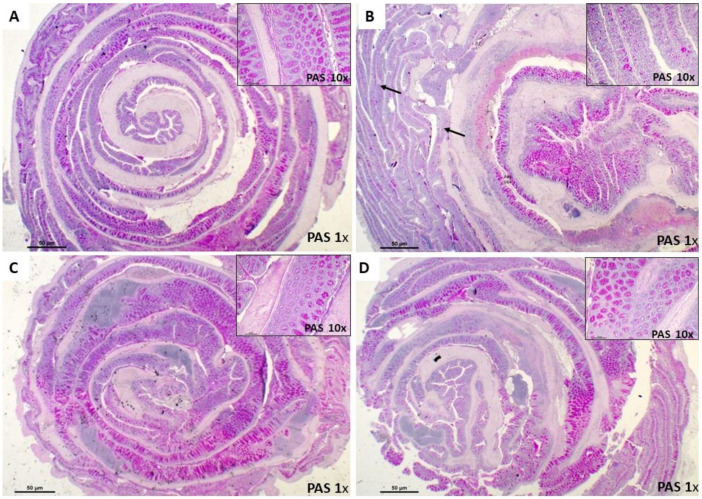
Preventive and curative effect of polyphenolic maqui extract (Ach) on the integrity of the colonic mucosa in a Crohn’s disease (CD) murine model. Representative PAS staining for goblet cells in the rolled entire large intestines (1× and 10× magnification) of each group. (**A**) Control Group: strong PAS+ staining for glycoproteins, including mucins, showing functional colonic mucosa and crypts arranged in the usual position. (**B**) CD Group: poor staining with PAS in the middle and proximal area (arrows) (**C**,**D**) Curative Group and Preventive Group: PAS+ staining goblet cells distribution similar to the Control Group.

**Figure 5 nutrients-12-01752-f005:**
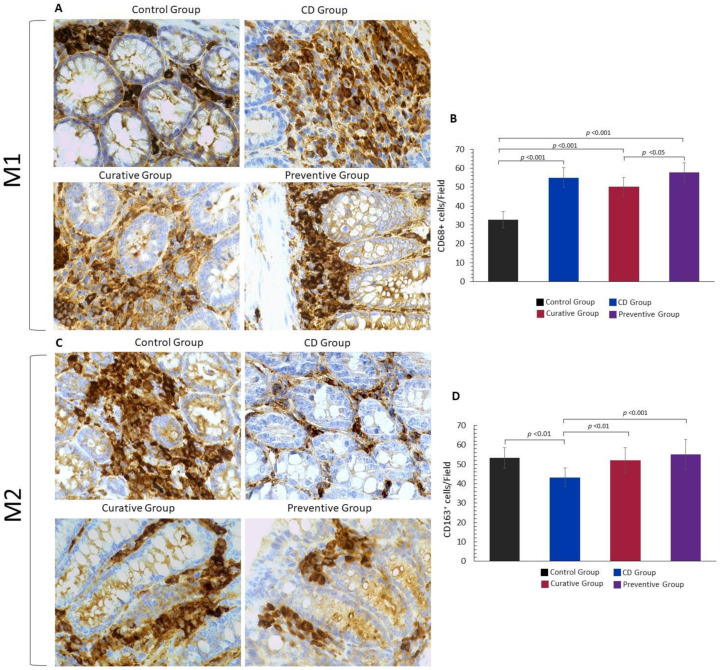
Polyphenolic maqui extract (Ach) restored M2 macrophage phenotypes in colonic tissue**.** (**A**) Immuno-histochemistry for CD68^+^ macrophages (M1) in the large intestine from all groups. (**B**) Number of CD68+ cells counted from ten high-power fields in the colonic tissue of each group. (**C**) Immuno-histochemistry for CD163^+^ macrophages (M2) in the large intestine from all groups. (**D**) Number of CD163^+^ cells counted from ten high-power fields in the colonic tissue of each group. All immune-histochemical images are shown at 40× magnification. Values represent mean ± SEM.

**Figure 6 nutrients-12-01752-f006:**
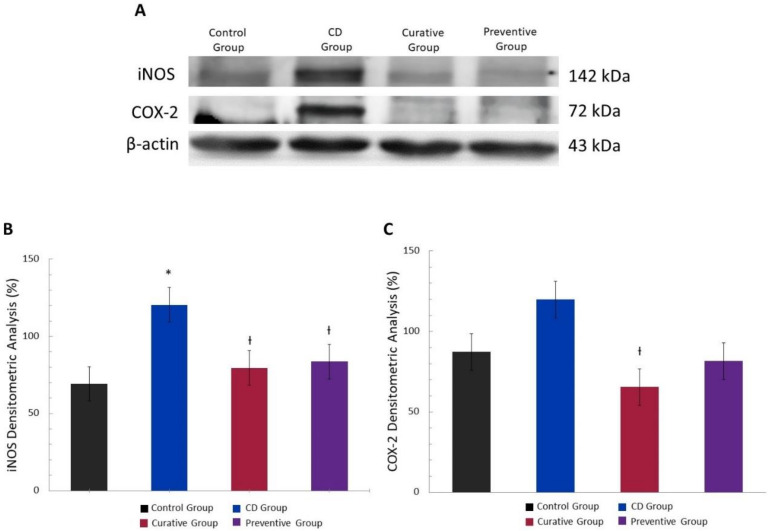
Anti-inflammatory effects of polyphenolic maqui extract (Ach) in the colon of mice from Crohn’s disease (CD) model. (**A**) Representative Western blot images of inducible nitric oxide synthase (iNOS) and cyclooxygenase-2 (COX-2) pro-inflammatory proteins of each group. β-actin was used as an equal loading control for normalization. (**B**) Densitometric analysis of iNOS. (**C**) Densitometric analysis of COX-2. Values represent mean ± SEM; * Statistical significance *p* < 0.05 compared to Control Group; Ɨ *p* < 0.05 compared to CD group.

**Figure 7 nutrients-12-01752-f007:**
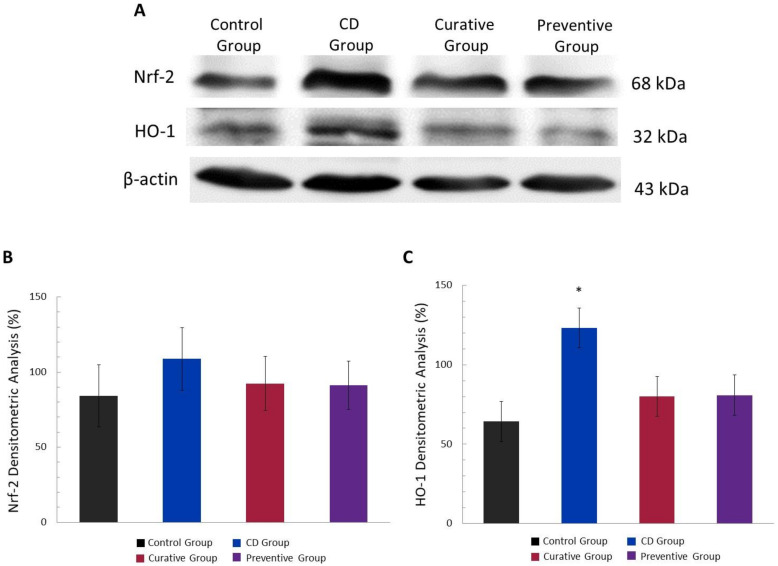
Antioxidant effect of polyphenolic maqui extract (Ach) in the colon of a TNBS-induced acute Crohn’s disease (CD) model. (**A**) Representative Western blot images of the nuclear factor (erythroid-derived 2)-like 2 (Nrf-2) and hemoxygenase-1 (HO-1) antioxidant proteins of each group. β-actin was used as an equal loading control for normalization. (**B**) Densitometric analysis of Nrf-2. (**C**) Densitometric analysis of HO-1. Values represent mean ± SEM; * Statistical significance *p* < 0.05 compared to Control Group.

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
