# Peer review of "Polyphenolic Maqui Extract as a Potential Nutraceutical to Treat TNBS-Induced Crohn’s Disease by the Regulation of Antioxidant and Anti-Inflammatory Pathways"

_nutrients, 2020, doi:10.3390/nu12061752_

Round 1
Reviewer 1 Report
Dear Author,
the manuscript resulted well written and clear for the reader. in my opinion, the experimental model is well designed and the results well described. the discuss is well addressed to the obtained results, however some informations should be added as indicated below:
- how the author have choose the amount of Ach to give to mice? preliminary experiment? please explain.
- have the authors evaluated the in vitro antioxidant capacity of Ach?
- the polyphenol content in Ach should be indicate to provide to the reader an idea of polyphenols provide to the animals.
- other point that should be discussed is relative to Nrf2. The authors have analyzed the expression of Nrf2 without consider its phosphorilated form. They found no significant changes in Nrf2 expression among experimental group, although they state that observed an over expression (see discussion lines 431-433 page 15). I suggest to consider the phosphorilated form of Nfr2 to confirm their hypothesis that antioxidant capacity of Ach play a key role in the improvement of TNBS animal model.
- the role of other antioxidant enzime regulated by Nrf2 could be involved. I suggest the authos to discuss also this point.
Author Response
Thank you very much for revising my manuscript entitled "Polyphenolic maqui extract as a potential nutraceutical to treat TNBS-induced Crohn's disease by regulation of antioxidant and anti-Inflammatory pathways ".
Here, we detail the changes made after answering your questions and suggestions. We added new sentences to the new version of the paper, they have been highlighted in red in this letter and with Track Changes function in the main document. Moreover, we made a few small corrections to the English.
Responses:
- How the author have choose the amount of Ach to give to mice? preliminary experiment? please explain.
Answer: The treatment dose of Ach was chosen according to reported previously. Our group did a bibliographic search between polyphenolic extract from berries and other fruit, herbal or natural source and its use in animal model of intestinal disease. Given that to date there are no studies regarding treatment with maqui in Crohn's disease and only one recently published study (Zhou. et al 2019. Maqui berry exhibited therapeutic effects against DSS-induced ulcerative colitis in C57BL/6 mice) has used maqui extract in ulcerative colitis at concentrations 50, 100 and 200 mg / kg (data published after carrying out our experiments), our choice was based on what was previously known.
The following sentence has been added to the paper: “Additionally, the treatment dose with Ach was chosen according to previously published in relation with polyphenols from natural source and treatment in the intestinal affections in experimental animal [40, 41]. However, the most studies shown positive effects against inflammation and clinical parameters using higher concentrations than our study [42, 43].”, Pag. 14, Line 378 - 382
- Have the authors evaluated the in vitro antioxidant capacity of Ach?
Answer: A fluorescent 2',7'-dichlorofluorescein diacetate (DCFH-DA) assay was performed to determine antioxidant capacity of our extract. However, the results of this parameter and others related trials (in vitro assays) are not indicated in the manuscript because they will be published in an another paper focused on chemistry properties, specifically in relation to the new methodology of polyphenol total extraction from Aristotelia chilensis, and it’s essential to add in that paper the antioxidant capacity and total polyphenol content. This paper has been already sent to a different journal.
The following sentence has been added to the paper: “…antioxidant effect evaluated through the intracellular ROS concentration using the DCFH-DA assay, …” Pag. 3, Line 95
- The polyphenol content in Ach should be indicate to provide to the reader an idea of polyphenols provide to the animals.
Answer: The value of this parameter has been added in Pag. 14, line 363-364, as follow:
“…has a total polyphenolic content of 39.02 mg/g of polyphenols (expressed as gallic acid) and…”
- Other point that should be discussed is relative to Nrf2. The authors have analyzed the expression of Nrf2 without consider its phosphorilated form. They found no significant changes in Nrf2 expression among experimental group, although they state that observed an over expression (see discussion lines 431-433 page 15). I suggest to consider the phosphorilated form of Nfr2 to confirm their hypothesis that antioxidant capacity of Ach play a key role in the improvement of TNBS animal model.
Answer: We have extensive experience in evaluating Nrf2 protein in different experimental animal models, including colitis (Ávila-Román et al. Anti-inflammatory effects of an oxylipin-containing lyophilised biomass from a microalga in a murine recurrent colitis model. Br J Nutr. 2016;116:2044-52), epidermal hyperplasia and UVB-Induced Skin Erythema (Rodríguez-Luna et al. Fucoxanthin-Containing Cream Prevents Epidermal Hyperplasia and UVB-Induced Skin Erythema in Mice. Mar Drugs. 2018;16:378) and UVB-Exposed Keratinocytes (Rodríguez-Luna et al. Fucoxanthin and Rosmarinic Acid Combination Has Anti-Inflammatory Effects through Regulation of NLRP3 Inflammasome in UVB-Exposed HaCaT Keratinocytes. Mar Drugs. 2019;17:451). In all these papers, we have determined Nrf-2 protein with an antibody from Santa Cruz Biotechnology (SC-722), which detect the protein at around 68 kDa. However, in the present paper, Ach did not induced a significant increase in Nrf-2 levels. Thus, in future studies it would be very interesting to deep into the mechanisms by which Ach exert its antioxidant activity, and evaluate phosphorilated Nfr2 (detected at 100 kDa) as well as other antioxidants enzymes regulated by Nrf-2 (ferritin, superoxide dismutase (SOD), peroxiredoxin-1 (PRDX1), gluthathione S-tranferases (GSTs)) .
By the way, we corrected the commercial references of two antibodies, Nrf-2 and HO-1 (Pag. 6, Line 207-208), we corrected the number of molecular weight observed of Nrf2 (68 KDa) at the top of the figure 7 in Pag 13. Line 341 and we added the following sentences: “The presence of OS leads to phosphorylation of serine/threonine residues in Nrf-2, to dissociate from Keap1 and translocation…” Pag 2. Line 75-76
- The role of other antioxidant enzyme regulated by Nrf2 could be involved. I suggest the authors to discuss also this point.
Answer: Thank you for your suggestion. We included in the manuscript other antioxidant enzyme regulated by Nrf-2, as follow:
“… and the consecutive regulation of the expression of numerous cytoprotective genes encoding antioxidant enzymes such as ferritin, superoxide dismutase (SOD), peroxiredoxin-1 (PRDX1), gluthathione S-tranferases (GSTs) [13, 55], and also its downstream target…”. Pag. 15, line 443-445
“… and also other antioxidants enzymes regulated by Nrf-2 could be involved during the first phase of oxidative and inflammation injury.” Pag. 16, line 455 - 456
Reviewer 2 Report
This manuscript investigated the effect of polyphenolic maqui extract as a potential nutraceutical to treat TNBS-induced acute Crohn's disease (CD) in mice. The authors have shown that the extract improved body weight, colon length, macroscopic and microscopic damage including transmural inflammation, promoted macrophage polarisation to M2 phenotype, and increased anti-oxidant and anti-inflammatory responses. This manuscript is well written with recent and relevant literature citations, methodology and results are clearly presented. The authors gave interesting data to show the effect of polyphenolic maqui extract on acute CD in mice, including the molecular mechanisms. However, the following points need to be addressed.
Please see my comments below-
- The authors studied the effects of the maqui extract on modulation of M1 and M2 phenotypes. In the introduction the authors can discuss role of M1 and M2 pathways as possible IBD targets. Below is an useful review Current Status of M1 and M2 Macrophages Pathway as Drug Targets for Inflammatory Bowel Disease, Arch Immunol Ther Exp (Warsz) 2020 Apr 1;68(2):10. PMID: 32239308.
- What is the source of the lyophilized powdered maqui fruit?
- Was the concentration of the polyphenols or anthocyanins in the extract detected? In the discussion the authors mentioned that delphinidin is the main polyphenol in the extract from references 19 and 34. Reference 34 shows the chromatogram of the maqui extract has higher delphinidin than cyanidin. Was the concentration of delphinidin detected in this study or is it similar to the referred paper?
- What is the reason behind the selection of the dose of 50 mg/kg/day of the maqui extract? What would be the equivalent amount of fresh maqui fruit for this dose? Is it relevant to human dose physiologically?
- The authors mention in the introduction that the study is on acute-phase CD animal model caused by TNBS. The section 2.3, figure 1 legend, results-line 230 mention ‘induction of colitis’, ‘colitis was induced’ and ‘colitis induction’ however the figure 1 legend also mentions ‘CD induction’ due to TNBS. Did the authors induce colitis in mice having CD features of human or did they induce only CD in mice because TNBS is usually to induce colitis in mice? Any literature citation for the current protocol of TNBS-induced acute phase CD in the methods section or was it developed by the authors?
- Were stool characteristics diarrhoea, rectal bleeding observed in the monitoring process of clinical parameters during the experiment?
- The authors can consider to discuss the possible role of gut microbiota and metabolites of polyphenols in mediating these effects of maqui extract.
Minor comments
- ‘TNBC-Induced Crohn's Disease’ is mentioned in the manuscript title. Is it TNBS?
- ‘TNBS-induced EC’ in section 3.2, line 235. Please expand EC.
- Please check the format of the references, esp the journal titles.
Author Response
Thank you very much for revising my manuscript entitled "Polyphenolic maqui extract as a potential nutraceutical to treat TNBS-induced Crohn's disease by regulation of antioxidant and anti-Inflammatory pathways ".
Here, we detail the changes made after answering your questions and suggestions. We added new sentences to the new version of the paper, they have been highlighted in red in this letter and with Track Changes function in the main document. Moreover, we made a few small corrections to the english.
Responses:
The authors studied the effects of the maqui extract on modulation of M1 and M2 phenotypes. In the introduction the authors can discuss role of M1 and M2 pathways as possible IBD targets. Below is an useful review Current Status of M1 and M2 Macrophages Pathway as Drug Targets for Inflammatory Bowel Disease, Arch Immunol Ther Exp (Warsz) 2020 Apr 1;68(2):10. PMID: 32239308.
Answer: Thank you for your suggestion. The reference was added in the Introduction, as follow:
“The balance between inflammatory M1 and anti-inflammatory M2 cells could determinate the disease progress and therefore the factors implicated in the disruption of the balance toward an increasing of the M2 macrophages cells could offer unique approaches for future management of IBD [9].” Pag. 2, Line 64-66.
- What is the source of the lyophilized powdered maqui fruit?
Answer: Our polyphenolic extract was obtained from lyophilized powdered wild maqui fruit (seed and pulp) from a packaged and commercialized product from Isla Natura de Chile®.
The following sentences has been added to the paper: “… lyophilized powdered wild maqui fruit (seed and pulp) from a packaged and commercialized product (Isla Natura de Chile®). 250 ml of MeOH/H+ (0.1% HCl) at pH 1 were added to the sample, …” Pag.3, Line 109-110.
- Was the concentration of the polyphenols or anthocyanins in the extract detected? In the discussion the authors mentioned that delphinidin is the main polyphenol in the extract from references 19 and 34. Reference 34 shows the chromatogram of the maqui extract has higher delphinidin than cyanidin. Was the concentration of delphinidin detected in this study or is it similar to the referred paper?
Answer: The value of this parameter has been added in Pag. 14, line 363-364, as follow:
“…has a total polyphenolic content of 39.02 mg/g of polyphenols (expressed as gallic acid) and…”
In this work we have not performed HPLC, so we do not have the polyphenolic profile. Anyway, as Reviewer 2 as noticed, delphinidin is the main polyphenol in the extract according to other authors.
- What is the reason behind the selection of the dose of 50 mg/kg/day of the maqui extract? What would be the equivalent amount of fresh maqui fruit for this dose? Is it relevant to human dose physiologically?
Answer: The treatment dose of Ach was chosen according to reported previously. Our group did a bibliographic search between polyphenolic extract from berries and other fruit, herbal or natural source and its use in animal model of intestinal disease. Given that to date there are no studies regarding treatment with maqui in Crohn's disease and only one recently published study (Zhou. et al 2019. Maqui berry exhibited therapeutic effects against DSS-induced ulcerative colitis in C57BL/6 mice) has used maqui extract in ulcerative colitis at concentrations 50, 100 and 200 mg / kg (data published after carrying out our experiments), our choice was based on what was previously known.
The following sentence has been added to the paper: “Additionally, the treatment dose with Ach was chosen according to previously published in relation with polyphenols from natural source and treatment in the intestinal affections in experimental animal [40, 41]. However, the most studies shown positive effects against inflammation and clinical parameters using higher concentrations than our study [42, 43].”, Pag. 14, Line 378 - 382
In relation to the other questions, we added in the discussion the follow sentence: “The equivalent amount of fresh maqui fruit for the dose administered to the mice was 100 mg/day. This value, translated to a standard human weight, corresponds to 240 g of fresh fruit per day.” Pag. 14, Line 388-390
- The authors mention in the introduction that the study is on acute-phase CD animal model caused by TNBS. The section 2.3, figure 1 legend, results-line 230 mention ‘induction of colitis’, ‘colitis was induced’ and ‘colitis induction’ however the figure 1 legend also mentions ‘CD induction’ due to TNBS. Did the authors induce colitis in mice having CD features of human or did they induce only CD in mice because TNBS is usually to induce colitis in mice? Any literature citation for the current protocol of TNBS-induced acute phase CD in the methods section or was it developed by the authors?
Answer: According with the first question is important to specify that the TNBS induces similar histological features to Crohn's disease in humans such as transmural inflammation. The acute Crohn's disease corresponds to a single dose of TNBS and its characteristic inflammatory infiltration. The colitis is considered a colon inflammation and its course is typical of Crohn's disease with affection in large intestine (our experimental model) and in the ulcerative colitis.
Respect to the protocol of TNBS-induced CD, we added this sentence: “…and CD was induced according to the method reported by Li T.J. et al. [37], with modifications.” Pag.3, Line 131-132
- Were stool characteristics diarrhea, rectal bleeding observed in the monitoring process of clinical parameters during the experiment?
Answer: The following sentences have been added to the paper:
“….induced a decreased stool consistency, rectal bleeding and significative weight loss in untreated mice…” Pag.7, Line 232-233
“…of Ach enhanced clinical parameters (diarrhea and blooding), ameliorated…” Pag. 14, Line 386
- The authors can consider to discuss the possible role of gut microbiota and metabolites of polyphenols in mediating these effects of maqui extract.
The authors are totally agree with this comment, but that topic will approached in a near future project. Also, we have added a short paragraph in discussion regarding this topic: “Moreover, a recent relevant report showed that maqui extract could exert an important effect on the imbalance of microbiota in IBD [55].” Pag. 16, Line 456-456
Minor comments
- ‘TNBC-Induced Crohn's Disease’ is mentioned in the manuscript title. Is it TNBS?
Yes, it is. It has been corrected in the new version of the paper.
- ‘TNBS-induced EC’ in section 3.2, line 235. Please expand EC.
The word “EC” was a mistake derived from Spanish: Enfermedad de Crohn. The right sentence is: “TNBS-induced CD”. It has been added to the paper Pag.8, Line 268
- Please check the format of the references, esp the journal titles.
The format used for the references was apply by Endnote X9 and the MDPI style was used for the development of this paper. The references were automatically inserted by the program.